# Climatic controls on watershed reference evapotranspiration vary dramatically during the past 50 years in southern China

Mengsheng Qin<sup>1</sup>, Lu Hao<sup>1</sup>, Lei Sun<sup>1</sup>, Yongqiang Liu<sup>2</sup>, and Ge Sun<sup>3</sup>

<sup>1</sup>Key Laboratory of Meteorological Disaster, Ministry of Education (KLME) / Joint International

5 Research Laboratory of Climate and Environment Change (ILCEC) / Collaborative Innovation Center on Forecast and Evaluation of Meteorological Disasters (CIC-FEMD) / Jiangsu Key Laboratory of Agricultural Meteorology, Nanjing University of Information Science & Technology, Nanjing 210044, China

<sup>2</sup>Center for Forest Disturbance Science, Southern Research Station, USDA Forest Service, Athens, GA

10 30602, USA

<sup>3</sup>Eastern Forest Environmental Threat Assessment Center, Southern Research Station, USDA Forest Service, Raleigh, NC 27606, USA

Correspondence to: Lu Hao (<u>hl\_haolu@163.com</u>); Ge Sun (<u>gesun@fs.fed.us</u>)

## 20 Abstract

Reference evapotranspiration (ETo) is an important hydrometeorological term widely used in water resource management, hydrological modeling, and understanding and projecting the hydrological effects of future climate change and land use change. Identifying the individual climatic controls on ETo helps better understand the processes of global climatic change impacts on local water resources and also simplify modeling efforts to predict actual evapotranspiration. We conducted a case study on the 25 Qinhuai River Basin (QRB), a watershed dominated by a humid subtropical climate and mixed land uses in southern China. Long term (1961–2012) daily meteorological data at six weather stations across the watershed were used to estimate ETo by the FAO-56 Penman-Monteith model. The seasonal and annual trends of ETo were examined using the Mann-Kendall nonparametric test. The individual 30 contributions from each meteorological variable were quantified by a detrending method. The results showed that basin-wide annual ETo had a decreasing trend during 1961–1987 due to decreased wind speed (WS), solar radiation ( $R_s$ ), vapor pressure deficit (VPD), and increased relative humidity (RH). These variables had different magnitudes of contribution to the ETo trend in different seasons examined during 1961–1987. However, during 1988–2012, both seasonal and annual ETo showed an increasing trend, mainly due to increased VPD and decreased RH and, to lesser extent, to decreased absolute 35 humidity (AH) and a rising air temperature. We show that the key climatic controls on ETo have dramatically shifted as a result of global climate change during the past five decades. Now the

40

atmospheric demand, instead of air temperature alone, is a major control on ETo. Thus, we conclude that accurately predicting current and future ETo and hydrological change under a changing climate must consider changes in VPD (i.e., air humidity and temperature) in the study region. Water resource management in the study basin must consider the increasing trend of ETo to meet the associated increasing water demand for irrigation agriculture and domestic water uses.

# **1** Introduction

In the past three decades, dramatic climate change and human activities have altered global hydrological cycles including the evapotranspiration (ET) processes (Xu et al., 2015; Zalewski, 2000), resulting in a series of environmental and socio-economic impacts (Roderick et al., 2002). Indeed, ET is a key component of water and energy balances, an important topic in modern ecohydrological, meteorological, agricultural, and ecological studies (Yang et al., 2012; Zhao et al., 2014). Reference ET (ETo) which is defined as the rate of ET over a hypothetical underlying surface with fixed parameters is an essential component in estimating actual ET (Liu et al., 2017), hydrological modeling, and projecting the impact of changing weather conditions on water supply (Allen et al., 1998; Liu et al., 2010). McMahon et al. (2013) provided summary of techniques to estimate reference ET and suggested that the FAO–56 Penman–Monteith model was widely adopted and preferred in humid regions. Quantifying the individual role of climatic factors in controlling ETo is important for understanding the influence of climatic change on hydrologic processes in terrestrial ecosystems (Fan and Thomas, 2012; Chen et al.,

70

2006). These investigations have remarkable theoretical and practical significance for understanding watershed hydrological cycle and effective use of farming water resources (Yang et al., 2014; Xu et al., 2015).

Worldwide, numerous studies have examined the trends of ETo in many different regions but with
different findings. Significant decreasing trends of ETo have been found in several river basins such as
the Platte River Basin in central Nebraska in the USA during 1893–2007 (Irmak et al., 2012), the Tons
River Basin, central India during 1969–2008 (Darshana et al., 2013), and China's Qinghai-Tibetan
plateau during 1971–2004 (Zhang et al., 2009) and Yunnan province during 1961–2004 (Fan et al., 2013). These studies showed that decreased ETo occurred in some regions despite the global
temperature has increased by 0.13 °C per decade in the last 50 years (IPCC, 2007).

The variable climatic controls on ETo have been examined under different climatic regimes. Jhajharia et al. (2012) found the decreased WS and declined net radiation overwhelmed the effect of increased air temperature, causing the decreased ETo in a humid region in northeast India during 1978–2002. In western Iran, the increasing trend of ETo was mainly caused by a significant increase in air temperature during 1966–2005 (Tabari et al., 2011a). No changes in ETo were found from 1964 to 1998 in Bet Dagan on Israel's central coastal plain because the effects of rising aerodynamic term were

In China, similar climate attribution studies have been conducted in different regions. Chen et al.

counterbalanced by the decreasing radiation term (Cohen et al., 2002).

(2011) proposed that the decrease in ETo in the Sichuan Basin of southwestern China was mainly due to the decreased sunshine hours. Zhang et al. (2010) found the combined effects of decreased WS and 75 sunshine hours offset the impact of increased air temperature, and then caused the decreased ETo in northeast China. Yang et al. (2014) proposed that increased maximum temperature was the main reason for the rise of ETo in Taohe River basin in northwest China during 1981–2010.

Our review of literatures suggests that (i) few studies have been carried out in the humid region of

- southern China, (ii) both increasing and decreasing trends of ETo were detected in different regions of 80 China and elsewhere, and (iii) not only air temperature affected changes of ETo but also other meteorological variables including wind speed (WS), solar radiation (R<sub>s</sub>), relative humidity (RH), absolute humidity (AH), and vapor pressure deficit (VPD) affected ETo to different extent. Therefore, there is a need to study the long term trend of ETo and comprehensive influence of various climatic factors to ETo in humid regions of southern China.

The Oinhuai River Basin (ORB) used for this case study has a subtropical climate typical of the lower Yangtze River Delta region in China. The region is rapidly developing and has been facing more environmental challenges such as land subsidence, water pollution, flooding, and urban heat islands (UHIs) (Liu et al., 2013; Zhao et al., 2014; Zhou et al., 2014). The QRB includes several cities with a population over 8 million and water demand for drinking, irrigation and industry use has placed great pressure on water resources management in this "water rich" region. Our previous study (Hao et al.,

2015) suggests that rapid urbanization by converting paddy fields and water withdrawal for large-scale irrigation in this basin have dramatically altered the watershed hydrology through changes in ET and surface water. A further understanding of the climatic control on ETo could provide valuable information to understand watershed hydrological processes and projecting the impacts of climatic change and land use change on water resources in this basin.

Based on previous studies on ETo in the humid region, we proposed two hypotheses to guide our research: (i) ETo has significantly increased during the past 50 years in the QRB, (ii) In addition to the increased air temperature, other meteorological variables have influenced ETo variation to a different

degree. In this study, we aimed to: (i) calculate the ETo over the QRB for the period of 1961–2012 by the FAO–56 Penman–Monteith model, (ii) determine the trends of ETo and eight key meteorological variables with the Mann-Kendall (MK) nonparametric test, and (iii) identify the dominant climatic factors in changing ETo at seasonal and annual scales by a detrending method.

## 2 Material and methods

## 105 **2.1 Study area and databases**

The QRB is located in the southwest of Jiangsu province  $(118^{\circ}39'-119^{\circ}19' \text{ E}, 31^{\circ}34'-32^{\circ}10' \text{ N})$  (Fig.1) and possesses an area of 2617 km<sup>2</sup> including Nanjing, Lishui, and Jurong cities. The land use is dominated by paddy rice field and dry cropland (60 %). The watershed has a flat topography with

elevation ranging from 0 to 412 m. The QRB has experienced a dramatic urbanization during the past

several decades (Du et al., 2012). By 2013, the built-up land has expanded to nearly 1/4 of the whole basin (Fig. 1). Irrigated paddy field covers nearly 35 % of the basin representing the dominant land use. Daily meteorological data from six standard weather stations in and around the QRB from 1961 to 2012 were provided by the China Meteorological Data Sharing Service System and Jiangsu Weather Bureau. Necessary variables to estimate ETo included WS (m s<sup>-1</sup>); RH (%); sunshine duration (n, h);
daily mean temperature (T<sub>mean</sub>, °C); daily maximum and minimum temperature (T<sub>max</sub> and T<sub>min</sub>, °C). Data from Jiangning station was available before 2007. The QRB is dominated by paddy rice field and

the rice growing season is between May and October when flood irrigation is needed (Hao et al., 2015).

Accordingly, besides four seasons and the annual scale, we also examined the rice growing season as the sixth study period.

## 120 2.2 FAO-56 Penman-Monteith model for estimating ETo

The FAO Penman–Monteith (P–M) model has been widely used to estimate ETo and applicable to humid conditions (Allen et al., 1998; McMahon et al., 2013). This model can be expressed:

$$ET_{0} \frac{0.408\Delta(R_{n}-G) + \gamma \frac{900}{T+273} U_{2}(e_{s}-e_{a})}{\Delta + \gamma(1+0.34U_{2})}$$
(1)

where  $\Delta$  is the slope of the saturated vapor pressure curve (kPa  $^{\circ}C^{-1}$ ),  $R_n$  the net radiation (MJ m<sup>-2</sup> 125 day<sup>-1</sup>), G the soil heat flux density (MJ m<sup>-2</sup> day<sup>-1</sup>) (zero on the daily scale), T the mean daily air

temperature (°C),  $U_2$  the mean daily wind speed at 2 m height (m s<sup>-1</sup>),  $e_s$  the saturated vapor pressure (kPa),  $e_a$  the actual vapor pressure (kPa),  $e_s - e_a$  the Vapor pressure deficit (kPa), and  $\gamma$  the psychrometric constant (kPa °C<sup>-1</sup>).

Key radiation part in P–M model was estimated by following equations:

$$130 \quad R_n = R_{ns} - R_{nl} \tag{2}$$

$$R_{ns} = (1 - \alpha)R_s \tag{3}$$

$$R_{s} = \left(a + b\frac{n}{N}\right)R_{a} \tag{4}$$

$$R_{nl} = \sigma \left(\frac{T_{\max,k}^4 + T_{\min,k}^4}{2}\right) \left(0.34 - 0.14\sqrt{e_a}\right) (1.35\frac{R_s}{R_{so}} - 0.35)$$
(5)

where  $R_{ns}$  and  $R_{nl}$  are the incoming net shortwave and outgoing net longwave radiation (MJ m<sup>-2</sup> d<sup>-1</sup>),  $\alpha$ 135 the albedo fixed as 0.23,  $R_s$  the solar radiation (MJ m<sup>-2</sup> d<sup>-1</sup>), n and N the actual and maximum possible sunshine hours (h) respectively,  $R_a$  extraterrestrial radiation (MJ m<sup>-2</sup> d<sup>-1</sup>),  $\sigma$  the Stefan-Boltzmann (4.903×10<sup>-9</sup> MJ m<sup>-2</sup> d<sup>-1</sup>),  $T_{max,k}$  and  $T_{min,k}$  the maximum and minimum absolute temperature within 24 hours (K),  $R_{so}$  the clear sky solar radiation (MJ m<sup>-2</sup> d<sup>-1</sup>), a and b the empirical coefficients of 0.25 and 0.5 (Allen et al., 1998).

## 140 The VPD (kPa) is calculated as (Allen et al., 1998):

$$VPD = e_s - e_a \tag{6}$$

where  $e_s$  (kPa) can be calculated from  $T_{max}$  and  $T_{min}$ :

$$e_{s} = \frac{e^{0}(T_{max}) + e^{0}(T_{min})}{2}$$
(7)

$$e^{0}(T_{\text{max}}) = 0.6108 \exp\left[\frac{17.27T_{\text{max}}}{T_{\text{max}} + 273.3}\right]$$
(8)

$$e^{0}(T_{\min}) = 0.6108 \exp\left[\frac{17.27T_{\min}}{T_{\min}+273.3}\right]$$
 (9)

Finally, we computed  $e_a$  (kPa) with the  $e_s$  and measured daily mean relative humidity (RH):

$$\mathbf{e}_{\mathbf{a}} = \mathbf{e}_{\mathbf{s}} * RH \tag{10}$$

Absolute humidity (AH, g m<sup>-3</sup>) was calculated with  $e_a$ :

$$AH = c \frac{e_a}{T}$$
(11)

where c is a constant with 217 (Xie et al., 2014).

#### 2.3 Mann-Kendall (MK) test

The nonparametric MK test (Mann 1945; Kendall 1975) was applied to analyze the trends of seasonal and annual ETo in many hydrological studies (Li et al., 2013; Liu et al., 2010; Fan et al., 2016). The statistic *S* is calculated as

$$S = \sum_{k=1}^{n-1} \sum_{j=k+1}^{n} sgn(X_j - X_k)$$
 (12)

where  $X_j$  represents the sequential data values, *n* is the number of the dataset, and

$$sgn(X_{j} - X_{k}) = \begin{cases} 1 & if X_{j} - X_{k} > 0\\ 0 & if X_{j} - X_{k} = 0\\ -1 & if X_{j} - X_{k} 

statistic S can be calculated by:

$$Var(S) = \frac{1}{18} \left[ n(n-1)(2n+5) - \sum_{w=1}^{v} t_p \left( t_w - 1 \right) (2t_w + 5) \right]$$
 (14)

where v is the number of tied groups and  $t_w$  is number of data values in wth group.

The standard test statistic (Z) is:

$$Z = \begin{cases} \frac{s-1}{\sqrt{Var(s)}} & \text{if } s > 0\\ 0 & \text{if } s = 0\\ \frac{s+1}{\sqrt{Var(s)}} & \text{if } s < 0 \end{cases}$$
(15)

The null hypothesis *H0* is rejected when  $|Z| > Z_{1-\alpha/2}$ , where  $Z_{1-\alpha/2}$  is the standard normal deviates.

## 165 **2.4 Theil-Sen's estimator**

Theil-Sen's estimator method was used to estimate magnitudes of ETo trends (Sen 1968; Hirsh 1982)

$$\beta = Median\left(\frac{X_j - X_k}{j - k}\right), 1 < k < j < n$$
(16)

where  $\beta$  is the estimated magnitude of slopes of ETo trends.  $\beta > 0$  represents an increasing trend;  $\beta 

variables to make them as stationary dataset, (ii) recalculating ETo with one detrended variable while
keeping the other variables unchanged, and (iii) comparing the recalculated ETo with original ETo. The contribution of changes in climatic factors to changes in ETo could be quantified by an evaluating indicator *R*:

$$R = \sum_{i=1}^{n} \frac{(ET_{o}^{o} - ET_{o}^{R})}{ET_{o}^{o}i}$$
(17)

where  $\text{ET}^{0}_{o}$  and  $\text{ET}^{R}_{o}$  are original and recalculated ETo respectively, n characterize the length of the data

180 set. R > 0 denotes the change of this climatic factor has positive effects to the changes in ETo; R