# Peer review of "Climatic controls on watershed reference evapotranspiration vary dramatically during the past 50 years in southern China"

_Hydrology and Earth System Sciences, 2017_

## Referee Comment (RC1) · Anonymous Referee #1 · 23 May 2017

This paper discusses the drivers of reference crop evapotranspiration (ETo), as specifically defined in FAO56 (Allen et al., 1956), for the humid Qinhuai River basin in eastern China. The authors applied the Mann-Kendall test, the Theil-Sen estimator and a detrending approach to assess trends. Based on analyses of data from six meteorological stations extending for 1961 to 2012, the authors observed a decrease in ETo over the period from 1961 to 1987 followed by an increase to the end of the period. They concluded that an increase in irrigation water demand for paddy rice crops is expected in the future.

General comments This paper is not a theoretical analysis but rather an application of standard procedures to a region in China where the specific procedures have not been

applied. Given the nature of the contribution, the paper is excessively long. Neverthe-less, I enjoyed reading the discussion in Section 4. The conclusions are supported by the analyses and the interpretation of results. I make the following recommendations. 1. Section 2.2 is well known material, and except for equation (1), the remaining material should be included as Supplementary Material. 2. I'm happy for Sections 2.3, 2.4 and 2.5 to be retained. 3. The results discussed in Section 3 need to be reduced to a small number of key observations. The current detail in Section 3 is unnecessary and very difficult to follow. All the information is shown clearly in the tables and figures which should be retained.

Specific comments 1. Ls48-50: Because the authors have adopted FAO56 reference crop equation (Allen et al. 1998) as the basis of their analysis, they need to be specific about the definition of a reference crop. A reference crop is defined as "A hypothetical reference crop with assumed crop height of 0.12 m, a fixed surface resistance of 70 s m-1 and an albedo of 0.23" (Allen et al., 1998, page 23). Quoting a secondary reference like Liu et al. (2017) is inappropriate especially as the definition is not specifically correct. 2. L49: It is not necessary to have estimates of ETo to determine actual ET. Actual ET is measured directly or through a water balance procedure where in some applications ETo is an input. 3. Ls94-96: I think this statement is optimistic. Understanding climate controls on actual ET is much more important. 4. Ls100-104: I suggest you mention Theil-Sen estimator and the De-trending method here. 5. L108: It would be helpful to report all the percentage areas for paddy rice fields, dryland cropping, woodland and urban areas. 6. L112: Please indicate whether there were missing data and, if so, how were the time series infilled. 7. L115: I assume daily temperature was not measured but the result of averaging maximum and minimum temperatures. Please clarify in the paper. 8. L116: What is the relevance of the statement "Data from Jiangning station was available before 2007" when the authors say nothing about the data from the other five stations? 9. L150: This section requires a concluding observation stating which variables will be discussed in the following analysis especially as the authors do not consider sunshine hours yet include solar radiation which is not a variable measured at the meteorological stations. 10. L184: Before you begin discussing the results there needs to be a short explanation of the choice of climate variables you plan to discuss, noting that observed data are available for only four variables – wind speed, sunshine hours, relative humidity and temperature, yet you are including in your discussion solar radiation which is unknown but, I assume, is estimated by Equation (4). You do not tell the reader how Ra is estimated which is required to estimate Rs. 11. L191: The authors have not said why they are discussing Rs when it is not a measured variable. Why not discuss n (sunshine hours) for which data are available? 12. L195: I don't think the authors tell the reader that the ETo values used in the analysis are calculated values based on Equation (1). 13. L195: This section (3.1) would have been clearer to me if the mutation point (I think I'd rather use the terms 'change point' or 'turning point') was discussed initially and then follow with a discussion of trends. 14. Ls 364,365: Delete the sentence beginning "However". Because of the different periods examined, this sentence has little merit. 15. L418-421: This has not been discussed before. Delete. It is not a conclusion. 16. Table 1: Please replace all "-" with the values of the trend. Although not statistically significant together they will provide a more complete picture of the trends. 17. Figure 2d: Please plot with same scale on both axes. 18. Add to Figure 2 caption before (a): 'Bars in the figures represent the average based on the six sites and the 52 years of data.' 19. Caption to Figure 5: Replace "1987" with '2012'. And delete from "and 1988-2012 . . . (Tmin)"

Technical corrections L25: simply → simplifies L44: I suggest the authors do not use emotive terms like "dramatic". They are unnecessary. L52: Add 'a' between "provided" and "summary" L57: Suggest rephrase to '. . . and effective use of water resources in irrigation. . .' L65: has increased → increasing L79: Suggest rephrase to 'Our literature review suggests . . .' L81: Suggest rephrase to 'and (iii) changes to ETo were affected not only by air temperature but . . .' L97: Suggest rephrase to 'Based on previous studies of ETo in humid regions, . . .' Ls118-119: Suggest rephrase to 'Accordingly, data for six periods were analysed: the four seasons, the annual period and the rice growing period.' L127: "Vapor pressure deficit" → 'vapor pressure deficit (VPD)' Ls173-174:

Suggest rephrase to 'This method comprises three steps: (i) removing for each variable the trend to render the variable stationary, ...' L176: Suggest rephrase to '... in ETo can be quantified by evaluating...' L179: Characterize → is L179: "n" Would it not be better to use another letter here as 'n' is defined earlier as the number of sunshine hours? L181: More appropriately "R = 0" should be replaced by 'R ≈ 0' L182: lead → leads L188: "Table 2" should be 'Table 1' L191: Suggest replace "which was similar to" with 'noted also by' L202: Slops → Slopes L214: Suggest rephrase to 'Variations in RH ...' L215: State which decades L223,224: I don't understand the sentence beginning "ETo of..". Please rephrase. L226: Would it be correct to say 'In each of the four seasons, the mean values ...' Ls234-235: This maybe better expressed as 'The daily detrended values were aggregated to yield detrended seasonal and annual detrended values.' L235: Delete full stop after "Figure" and delete "obvious". This adjective is not necessary. L236: Suggest rephrase to ''...detrended variables at the annual time step' L238: drtended → detrended L243: Suggest rephrase to 'It is noted that ...'. L251: Replace "between annual original ETo and recalculated" with 'between the original annual ETo and the recalculated annual ETo...' L254: Delete "phenomenon". L257: Delete "Obviously,". L260: Replace "on the" with 'at the'. L265: Delete "only" and replace "and then" with 'which were' L272: On → At L277: Replace "In" with ' During the' L287: Add 'in the growing'. L294: phenomenon → feature L295: Delete "the" L301: Rephrase 'We found significant decreases in RH...'. L304: evapotranspiration → evaporation (One cannot have ET from an ocean as there is no vegetation.) L304: Rephrase "limited the evapotranspiration" with 'reduce relatively the evaporation'. L304,305: Rephrase "from oceans eventually" with 'from the oceans'. L306: was corresponding → corresponded L308: increase → increases L309: replace "cement surface" with 'hard surfaces' L310: city → cities. L314: Delete "tendency". L315: Replace "It is unclear about the causes of AH variation" with 'The causes of the AH variation are unclear'. L318: Replace "globally (Matsoukas et al., 2011). Matsoukas et al. (2011) suggested that the global" with 'globally by Matsoukas et al. (2011). They suggest that global ...' L320: Replace "of" with 'in'. L321: I assume Platte River Basin is in USA. Please say so. L322:

[Figure]

Add 'strongly' after "correlated" and delete "to a great extent". Also add a reference supporting the Platte River increased VPD. L323: Add after "was" 'found to be ' L323: on → in L323,324: Replace "1988-2012 and attributed to these reasons" with '1988-2012. The is attributed to two reasons:' L325: Delete "in QRB". L326: Add after "that" 'there was an' and replace "were found in vast majority" with 'during most'. L333: Add 'and to' before "vast" L338: Add 'the' before "annual" L339: Replace "at" with 'for the' L353: Replace "Different from these literatures" with 'In contrast' L354: Rephrase as follows 'driven not only by air' L354: Add 'by ' before "other" L355: Replace "periods. During 1961−1987," with 'periods, 1961-1987 and 1988-2012. During the first period, decreased. . .' L359: After "same" add 'result' L363: Replace "with the" with 'for a' L367: Replace "another word" with 'other words' L368: was → is L371: Replace "S" with 's' L375: Delete "water shed" L377: Delete "during the recent 30 years" L381: Delete "new" L391: L391,392: Replace "to water vapour and" with 'and hence' L392: Delete "has" L394: increase → rise L395: delete "then" L396: influence → influences

---

## Referee Comment (RC2) · Anonymous Referee #2 · 26 Jun 2017

In this manuscript, the authors present an analysis of daily meteorological data for 6 weather stations in (the near surroundings of) the Qinhuai river basin in in China. They derive evaporation data using the Penman-Monteith model and look for annual and seasonal trends. Subsequently, they analyse relations with various meteorological variables to identify their potential contributions to changes in evaporation values. One of the motivations for this study is that few such studies have been done for the humid region of southern China. This at the same time is an important limitation of the work: it's not clear what new insights are derived from analysis of data from only 6 weather stations within a limited geographical scope. In its present form and given the limited scope, I consider the manuscript unsuitable for publication in HESS. Following

are some general recommendations to improve the manuscript: - Dataset description: add information on instrumentation of the meteorological stations: what variables are measured, at what resolution, what sensors are used, what is the mean distance between stations, report data control procedures and data gaps, if any. - Equation (1): equal sign seems to be missing - On p12, l 199 it is stated that "spatially average wind speeds" were derived. Wind speeds are typically highly variable in space, so they cannot be simply interpolated across 10-20km distances. This needs more explanation or rather, stick to analysis of the individual data series per station. - Reporting both relative humidity and water vapour deficit seems superfluous, since 1 directly depends on the other - Conclusion: explain better what this study contributes to the general body of knowledge. Most of what is currently in the conclusions chapter are interpretations of the findings towards practical application. Consider moving this to the discussions chapter. - English language needs improvement throughout the manuscript

---

## Author Comment (AC1) · 16 Jul 2017

Dear Reviewer,

First of all, we appreciate your interest in this study and your insightful suggestion to make our conclusions more convincing. Accordingly, we provide following response point to point:

General comments: Section 2.2 is well-known material, and except for equation (1), the remaining material should be included as Supplementary Material.

[Figure]

Response: Yes, we have retained the FAO-56 Penman-Monteith and added the other equations into the Supplement Material.

I'm happy for Sections 2.3, 2.4 and 2.5 to be retained.

Response: We have deleted the original 2.3, 2.4 and 2.5 sections and briefly introduce the data analysis methods in new 2.3 section: Statistical analysis, and add the detailed steps of the methods to the Supplementary Material.

The results discussed in Section 3 need to be reduced to a small number of key observations. The current detail in Section 3 is unnecessary and very difficult to follow. All the information is shown clearly in the tables and figures which should be retained.

Response: Thank you for this valuable suggestion. To address the reviewer's concerns we have made two major efforts in this revision: We have deleted the complex expressions, revised the confusing and unclear technical details. We hope these changes can make the paper easier to understand. Some subtitles were added in this paper (3.1.1 Climatic characteristics over QRB, 3.1.2 Trends of climatic variables in 1961-1987 and 1988-2012, 3.3.3 Contributions of climatic variables to trends of ETo at both seasonal and annual scale).

Specific comments:

Ls48-50: Because the authors have adopted FAO56 reference crop equation (Allen et al. 1998) as the basis of their analysis, they need to be specific about the definition of a reference crop. A reference crop is defined as "A hypothetical reference crop with assumed crop height of 0.12 m, a fixed surface resistance of 70 s m-1 and an albedo of 0.23" (Allen et al., 1998, page 23). Quoting a secondary reference like Liu et al. (2017) is inappropriate especially as the definition is not specifically correct.

Response: Thank for the advice. We have quoted the definition of reference crop "A hypothetical reference crop with assumed crop height of 0.12 m, a fixed surface resistance of 70 s m-1 and an albedo of 0.23" (Allen et al., 1998)

[Figure]

L49: It is not necessary to have estimates of ETo to determine actual ET. Actual ET is measured directly or through a water balance procedure where in some applications ETo is an input.

Response: we agree. The sentence was revised "essential" to "important" in line 49.

Ls94-96: I think this statement is optimistic. Understanding climate controls on actual ET is much more important.

Response: Thanks for your excellent suggestions. We revised the sentence to better describe this point (line 94). The revised sentence is "A further understanding of the climatic control on ETo could help us better understand the climatic control on actual ET, thuscontribute to comprehend watershed hydrological processes and project the impacts of climatic change and land use change on water resources in this basin."

Ls100-104: I suggest you mention Theil-Sen estimator and the De-trending method here.

Response: we have mentioned these two methods here.

L108: It would be helpful to report all the percentage areas for paddy rice fields, dryland cropping, woodland and urban areas.

Response: We have added percentage areas of each land cover in Section 2.1as "The land use is dominated by paddy rice field and dry cropland which occupied for nearly 33.4% and 26.1% of the whole basin respectively. Woodland and urban areas occupied for 11.9 and 24.1% of this basin (Fig.1)."(line106)

L112: Please indicate whether there were missing data and, if so, how were the time series infilled.

Response: The missing data were limited in our study. Except for data in Jiangning station which were unavailable after 2007, the data in other five stations provided by China Meteorological Data Sharing Service System and Jiangsu Weather Bureau

was almost complete (missing daily data < 1%). These data were processed and the missing data were interpolated according to chapter 23 in this book "The Criterion of Surface Meteorological Observation, 2004, China Meteorological Administration".

L115: I assume daily temperature was not measured but the result of averaging maximum and minimum temperatures. Please clarify in the paper.

Response: Also according to section 20.4 in the book "The Criterion of Surface Meteorological Observation, 2004, China Meteorological Administration". The mean daily climatic variables such as relative humidity, temperature, wind speed were computed by averaging the observations at 02am, 08am, 02pm, 08pm.

L116: What is the relevance of the statement "Data from Jiangning station was available before 2007" when the authors say nothing about the data from the other five stations?

Response: The revised sentence is "Data from Jiangning station was only available prior to 2007, while the other five stations have almost complete data from 1961 to 2012. Therefore, ETo at Jiangning station during 2007-2012 was null." (line 116)

L150: This section requires a concluding observation stating which variables will be discussed in the following analysis especially as the authors do not consider sunshine hours yet include solar radiation which is not a variable measured at the meteorological stations.

Response: Thank you for this valuable suggestion. We have added the reasons for selecting these variables which are discussed in the last paragraph in section 2.3. The evapotranspiration process is determined by the amount of energy available to vaporize water and Rs is the largest energy source (Allen et al., 1998).

Besides sunshine hours, we must consider the locations of evaporating surface when evaluating impacts of radiometric term on ETo trends. Rs not only consider the sunshine hours and the location of weather station, but also consider the position of the

sun for 365 days throughout the year.

L184: Before you begin discussing the results there needs to be a short explanation of the choice of climate variables you plan to discuss, noting that observed data are available for only four variables – wind speed, sunshine hours, relative humidity and temperature, yet you are including in your discussion solar radiation which is unknown but, I assume, is estimated by Equation (4). You do not tell the reader how Ra is estimated which is required to estimate Rs.

Response: Yes, we have added the reasons for selecting these variables which are discussed in the last paragraph in section 2.3. We also have added the equation (5) which indicates that how the extraterrestrial radiation (Ra) was obtained into the supplementary material. Ra=1440/$\pi$ G_sc d_r $\omega$_s sin($\varphi$)sin($\delta$)+cos($\varphi$)cos($\delta$)sin($\omega$_s ) (5) Ra extraterrestrial radiation (MJ m$-$2 d$-$1), Gsc the solar constant (0.08232 MJ m$-$2 min$-$1), dr inverse relative distance from earth-to-sun, $\omega$_s the sunset hour angle (rad), $\varphi$ latitude (rad), $\delta$ solar declination(rad),

L191: The authors have not said why they are discussing Rs when it is not a measured variable. Why not discuss n (sunshine hours) for which data are available?

Response: Good suggestion on discussing Rs. We have answered this question in comment 9.

L195: I don't think the authors tell the reader that the ETo values used in the analysis are calculated values based on Equation (1).

Response: We have revised the sentence as "The MK test and linear regression both showed that the annual ETo estimated by FAO 56 P-M model over QRB has significantly decreased (p < 0.01) during 1961$-$1987, then significantly increased (p < 0.01) during 1988$-$2012 (Table 3 and Fig. 3f)."line195

L195: This section (3.1) would have been clearer to me if the mutation point (I think I'd rather use the terms 'change point' or 'turning point') was discussed initially and then

follow with a discussion of trends.

Response: I am sorry for the incorrect expression of "mutation point" and negligence of the change points. We have listed Table 1 reporting change points (tested by Cramer's test) for each variable. We added this table in the Supplementary material.

Table 1. Change points of climatic variables in Qinhuai River Basin during 1961-2012

Spring Summer Autumn Winter Growing season Annual Rs (MJ m$-2$ d$-1$) 1994 1984 1985 1993 1985 1987 WS (m s$-1$) 1993 1993* 1993* 1993* 1993* 1993* RH (%) 1986 1984* 1985* - 1987* 1987* AH (g m3) 1983 1984 - 1983 - 1983 VPD (kpa) 1984 1987* 1986* 1986 1987* 1987* Tmean ($\circ$C) 1986* 1986* 1986* 1986 1986* 1986* Tmax ($\circ$C) 1986* 1986* 1979* 1986 1986* 1986* Tmin ($\circ$C) 1986* 1986* 1986* 1980* 1986* 1986*

Ls 364,365: Delete the sentence beginning "However". Because of the different periods examined, this sentence has little merit.

Response: Done.

L418-421: This has not been discussed before. Delete. It is not a conclusion.

Response: Done.

Table 1: Please replace all "-" with the values of the trend. Although not statistically significant together they will provide a more complete picture of the trends.

Response: We have added the values of insignificant trends in Table 2.

Table 2. Trends of key meteorological variables in Qinhuai River Basin during 1961$-$1987 and 1988$-$2012 Periods Variables Spring Summer Autumn Winter Growing season Annual 1961$-$ 1987 Rs (MJ·m$-2$·d$-1$) 0.013 $-0.061$* 0.007 -0.021 -0.031 $-0.027$* WS (m·s$-1$) -0.031*** $-0.014$* -0.032*** -0.034*** $-0.025$*** -0.027*** RH (%) -0.103 0.104* 0.035 -0.03 0.071 0.033 AH (g m3) -0.012- -0.029 -0.012 -0.008 -0.013 -0.013 VPD (kpa) 0.001

- −0.005** -0.001 0.001 -0.003 -0.002 Tmean (°C) -0.009 -0.043* -0.01 -0.003 -0.015 -0.013 Tmax (°C) -0.02 −0.054** -0.017 -0.018 −0.026+ −0.029* Tmin (°C) -0.007 −0.029* -0.005 -0.007 -0.018 -0.011 1988− 2012 Rs (MJ m−2 d−1) 0.048+ −0.073+ −0.051* -0.02 −0.049+ -0.018 WS (m s−1) 0.002 0.001 0.002 0.004 -0.001 0.002 RH (%) −0.52*** −0.31*** −0.20* −0.28*** −0.33*** −0.38*** AH (g m3) -0.026 −0.046* -0.006 −0.024+ −0.030+ -0.025 VPD (kpa) 0.014*** 0.014*** 0.007*** 0.003* 0.013*** 0.010*** Tmean (°C) 0.079** 0.038+ 0.058** -0.014 0.045** 0.039* Tmax (°C) 0.114** 0.037+ 0.03 -0.028 0.048** 0.038+ Tmin (°C) 0.056+ 0.045* 0.084** 0.001 0.055*** 0.041*

Figure 2d: Please plot with same scale on both axes.

Response: Done.

Fig. 2. Basic meteorological information in Qinhuai River basin during 1961−2012. Bars in the figures represent the average based on the six sites and the 52 years of data. (a) Reference evapotranspiration (ETo) and Precipitation (P); (b) Relative humidity (RH) and Absolute humidity (AH); (c) Vapor pressure deficit (VPD) and Mean temperature (Tmean); and (d) Maximum temperature (Tmax) and Minimum temperature (Tmin).

Add to Figure 2 caption before (a): 'Bars in the figures represent the average based on the six sites and the 52 years of data.'

Response: We have revised.

Caption to Figure 5: Replace "1987" with '2012'. And delete from "and 1988-2012 . . . (Tmin)"

Response: We have revised

Technical correctionsïijŽ L25: simply → simplifies

Response: Modified

L44: I suggest the authors do not use emotive terms like "dramatic". They are unnecessary

Response: We deleted the word "dramatic", as well as the words like "extremely, obvious…"

L52: Add 'a' between "provided" and "summary"

Response: Modified

L57: Suggest rephrase to '. . . and effective use of water resources in irrigation. . .'

Response: Modified

L65: has increased → increasing

Response: Modified

L79: Suggest rephrase to 'Our literature review suggests . . .'

Response: Modified

L81: Suggest rephrase to 'and (iii) changes to ETo were affected not only by air temperature but . . .'

Response: Modified

L97: Suggest rephrase to 'Based on previous studies of ETo in humid regions, . . .'

Response: Modified

Ls118-119: Suggest rephrase to 'Accordingly, data for six periods were analysed: the four seasons, the annual period and the rice growing period.'

Response: Modified

L127: "Vapor pressure deficit" → 'vapor pressure deficit (VPD)'

Response: Modified

Ls173-174: Suggest rephrase to 'This method comprises three steps: (i) removing for each variable the trend to render the variable stationary, . . .'

Response: Modified.

L176: Suggest rephrase to '. . . in ETo can be quantified by evaluating. . .'

Response: Modified.

L179: Characterize → is

Response: Modified

L179: "n" Would it not be better to use another letter here as 'n' is defined earlier as the number of sunshine hours?

Response: Modified

L181: More appropriately "R = 0" should be replaced by 'R $\approx$ 0'

Response: Done

L182: lead → leads 210

Response: Done

L188: "Table 2" should be 'Table 1'

Response: Modified

L191: Suggest replace "which was similar to" with 'noted also by'

Response: Modified

L202: Slops → Slopes

Response: Modified

L214: Suggest rephrase to 'Variations in RH . . .'

Response: Modified

L215: State which decades

Response: we have stated in paper "in the past three decades"

L223,224: I don't understand the sentence beginning "ETo of..". Please rephrase.

Response: The sentence has been revised as "The multi-year average ETo in $1961-1987$ and $1988-2012$ were both close to 1000mm, 70% of which occurred in the growing season."

L226: Would it be correct to say 'In each of the four seasons, the mean values . . .'

Response: Modified

Ls234-235: This maybe better expressed as 'The daily detrended values were aggregated to yield detrended seasonal and annual detrended values.'

Response: Modified

L235: Delete full stop after "Figure" and delete "obvious". This adjective is not necessary.

Response: Modified

L236: Suggest rephrase to ''. . .detrended variables at the annual time step'

Response: Modified

L238: drtended → detrended

Response: Modified

L243: Suggest rephrase to 'It is noted that . . .'.

Response: Modified

L251: Replace "between annual original ETo and recalculated" with 'between the original annual ETo and the recalculated annual ETo. . .'

Response: Modified

L254: Delete "phenomenon".

Response: Modified

L257: Delete "Obviously,".

Response: Modified

L260: Replace "on the" with 'at the'.

Response: Modified

L265: Delete "only" and replace "and then" with 'which were'

Response: Modified

L272: On → At

Response: Modified

L277: Replace "In" with ' During the'

Response: Modified

L287: Add 'in the growing'. Response: Modified

L294: phenomenon → feature

Response: Modified

L295: Delete "the"

Response: Modified

L301: Rephrase 'We found significant decreases in RH...'.

Response: Modified

L304: evapotranspiration → evaporation (One cannot have ET from an ocean as there is no vegetation.)

Response: I am sorry for the negligence. We have modified.

L304: Rephrase "limited the evapotranspiration" with 'reduce relatively the evaporation'.

Response: Modified

L304,305: Rephrase "from oceans eventually" with 'from the oceans'.

Response: Modified

L306: was corresponding → corresponded

Response: Modified

L308: increase → increases

Response: Modified

L309: replace " cement surface" with 'hard surfaces'

Response: Modified

L310: city → cities.

Response: Modified

L314: Delete "tendency".

Response: Modified

L315: Replace "It is unclear about the causes of AH variation" with 'The causes of the AH variation are unclear'.

Response: Modified

L318: Replace "globally (Matsoukas et al., 2011). Matsoukas et al. (2011) suggested that the global" with 'globally by Matsoukas et al. (2011). They suggest that global . . .'

Response: Modified

L320: Replace "of" with 'in'.

Response: Modified

L321: I assume Platte River Basin is in USA. Please say so.

Response: Modified

L322: Add 'strongly' after "correlated" and delete "to a great extent". Also add a reference supporting the Platte River increased VPD.

Response: We have revised the sentence below this suggestion. We also added a reference (Irmak, S., Kabenge, I., Skaggs, K. E., and Mutiibwa, D.: Trend and magnitude of changes in climate variables and reference evapotranspiration over 116-yr period in the Platte River Basin, central Nebraska–USA, J. Hydrol., $420-421$, $228-244$, doi:10.1016/j.jhydrol.2011.12.006, 2012." To suppot the increased VPD in Platte River Baisn, central Nebraska – USA.

L323: Add after "was" 'found to be '

Response: Modified

L323: on $\rightarrow$ in

Response: Modified

L323,324: Replace "1988-2012 and attributed to these reasons" with '1988- 2012. The is attributed to two reasons:'

Response: Modified

L325: Delete "in QRB".

Response: Modified

L326: Add after "that" 'there was an' and replace "were found in vast majority" with 'during most'.

Response: Modified

L333: Add 'and to' before "vast"

Response: Modified

L338: Add 'the' before "annual"

Response: Modified

L339: Replace "at" with 'for the'

Response: Modified

L353: Replace "Different from these literatures" with 'In contrast'

Response: Modified

L354: Rephrase as follows 'driven not only by air'

Response: Modified

L354: Add 'by " before "other" Response: Modified

L355: Replace "periods. During $1961-1987$," with 'periods, 1961-1987 and 1988-2012. During the first period, decreased. . .'

Response: Modified

L359: After "same" add 'result'

Response: Modified

L363: Replace "with the" with 'for a'

Response: Modified

L367: Replace "another word" with 'other words'

Response: Modified

L368: was → is

Response: Modified

L371: Replace "S" with 's'

Response: Modified

L375: Delete "water shed"

Response: Modified

L377: Delete "during the recent 30 years"

Response: Modified

L381: Delete "new"

Response: Modified

L391,392: Replace "to water vapor and" with 'and hence'

Response: Modified

L392: Delete "has"

Response: Modified

L394: increase → rise

Response: Modified

L395: delete "then"

Response: Modified

L396: influence → influences

Response: Modified

Please also note the supplement to this comment:
https://www.hydrol-earth-syst-sci-discuss.net/hess-2017-241/hess-2017-241-AC1-supplement.pdf

[Figure]

**Fig. 1.** Fig. 2. Basic meteorological information in Qinhuai River basin during 1961−2012.

---

## Author Comment (AC2) · 16 Jul 2017

Dear Reviewer,

First of all, we appreciate your interest in this study and your insightful suggestion to make our conclusions more convincing. Accordingly, we provide following response point to point.

(1) In this manuscript, the authors present an analysis of daily meteorological data for 6 weather stations in (the near surroundings of) the Qinhuai river basin in China.

They derive evaporation data using the Penman-Monteith model and look for annual and seasonal trends. Subsequently, they analyze relations with various meteorological variables to identify their potential contributions to changes in evaporation values. One of the motivations for this study is that few such studies have been done for the humid region of southern China. This at the same time is an important limitation of the work: it's not clear what new insights are derived from analysis of data from only 6 weather stations within a limited geographical scope. In its present form and given the limited scope, I consider the manuscript unsuitable for publication in HESS.

Response:

The Qinhuai River Basin used for this case study has a subtropical climate typical of the lower Yangtze River Delta region (210,700 km2) in China. Working at a watershed scale, we had the opportunity to look at all aspects of the hydrometeorological changes. We have further clarified this motivation using one watershed to derive information for a bigger region.

This study focuses on climatic control on ETo over QRB, and results can provide information for the similar areas in the Yangtze River Delta. Using data from the 6 weather stations that cover the entire study basin, we indeed found an interesting new trend in ETo. Based on the results of previous studies, we initially thought that ETo in QRB would increase in the past 50 years as a result of the increasing temperature. However, we eventually found that ETo in QRB showed a significant decreasing trend in 1961-1987, then significantly increased in 1988-2012. Decreasing solar radiation, wind speed, VPD and increased RH were the main reasons for the decreasing ETo in 1961-1987. Since 1988, atmospheric demand (increased VPD and decreased RH), instead of air temperature alone, is the major control on the increased ETo. Thus, we conclude that accurately predicting current and future ETo and hydrological change under a changing climate must consider changes in VPD (i.e., air humidity and temperature) in the study region. Water resource management in the study basin must consider the increasing trend of ETo to meet the associated increasing water demand for irrigation

agriculture and domestic water uses.

(2) Dataset description: add information on instrumentation of the meteorological stations: what variables are measured, at what resolution, what sensors are used, what is the mean distance between stations, report data control procedures and data gaps

Response:

The daily data from six weather stations were provided by the China Meteorological Data Sharing Service System and Jiangsu Weather Bureau. All the observations of meteorological variables follow the standards edited by World Meteorological Organization (WMO). The measurement of temperature, relative humidity, surface wind speed and sunshine duration can be found in section 2, 4, 5 and 8 of the book "World Meteorological Organization Guide to Meteorological Instruments and Methods of Observation, 2006". According to section 20.4 in the book "The Criterion of Surface Meteorological Observation, 2004, China Meteorological Administration.", the mean daily climatic variables such as relative humidity, temperature, wind speed were computed by averaging the observations at 02 am, 08 am, 02 pm, 08 pm. For your kind suggestion, more detailed introduction to the instrumentation can be found in this book. I have displayed the linear distances between each station in Table 1 and add this table in Supplementary Material. Except for the data in Jiangning station which was available before 2007, the data in other five stations offered by China Meteorological Data Sharing Service System and Jiangsu Weather Bureau was almost complete (missing daily data < 1%). These data were processed and interpolated according to section 23 in this book "The Criterion of Surface Meteorological Observation, 2004, China Meteorological Administration".

Table 1 The linear distances between each station (km)

Stations Liuhe Pukou Nanjing Jiangning Lishui Jurong Liuhe - 42 35 44 81 56 Pukou 42 - 19 27 61 56 Nanjing 35 19 - 12 49 37 Jiangning 44 27 12 - 39 29 Lishui 81 61 49 39 - 32 Jurong 56 56 37 29 32 -

(3) Equation (1): equal sign seems to be missing

Response: Modified.

(4) On p12, l 199 it is stated that "spatially average wind speeds" were derived. Wind speeds are typically highly variable in space, so they cannot be simply interpolated across 10-20km distances. This needs more explanation or rather, stick to analysis of the individual data series per station.

Response:

Thank you for this valuable suggestion. I am sorry for the incorrect expression "spatially average wind speed". In this study, we evaluated the impacts of changes in climatic variables to ETo trends by using the station data, not the interpolated spatial data. We separately computed the recalculated ETo by using the detrended wind speed from each station. Due to space limitations, we evaluated the impacts of wind speed over QRB by comparing averaged recalculated ETo with detrended Wind speed and the original averaged ETo.

(5) Reporting both relative humidity and water vapor deficit seems superfluous, since 1 directly depends on the other –

Response:

Thank you for your suggestion. We would like to retain these two variables because they have different definitions and meanings to ecosystems and climate system. Relative humidity (%) represents the degree of moisture content of atmosphere and commonly used to analyze the impacts of air humidity to ETo trends. Vapor pressure deficit, determined as the difference between saturated vapor pressure and actual vapor pressure, is directly related to atmospheric demand for water (Novick et al, 2016). We have added the reasons for selecting these variables in the last paragraph in section 2.2.

(6) Conclusion: explain better what this study contributes to the general body of knowledge. Most of what is currently in the conclusions section are interpretations of the

findings towards practical application. Consider moving this to the discussions section.

Response:

Based on the Reviewer's suggestion, we added new findings in section 5 line 410: (1) The environment is getting drier over the Qinhuai River Basin in the humid southern China. It is unknown how this change may affect crops that are used to grow in a humid climate. (2) Atmospheric demand, instead of air temperature alone, is a major control on ETo. Predicting hydrological change under a changing climate must consider both air humidity and air temperature. (3) This study has important implications for watershed management in these paddy field-dominated regions, and similar humid regions, where actual water loss is mainly controlled by atmospheric demand.

We have modified the conclusion as follows:

This long-term study (1961−2012) shows that ETo over Qinhuai River Basin has changed significantly over the past 52 years: a decreasing trend during 1961−1987 and then an increasing trend during 1988−2012. Prior to 1987, decreased WS, Rs, VPD and increased relative humidity were responsible for the negative trends of ETo. The positive trends of ETo during 1988−2012 were mainly caused by effects of decreased relative humidity and increased VPD. The decreased absolute humidity and increased air temperature also contributed to the increased ETo to a lesser degree.

Generally speaking, our new findings in this study showed that the environment is getting drier over the Qinhuai River Basin in the humid southern China. It is unknown how this change may affect crops that are used to a humid climate Secondly, atmospheric demand, instead of air temperature alone, is a major control on ETo. Predicting hydrological change under a changing climate must consider both air humidity and air temperature. Our study also has important implications for watershed management in these paddy field-dominated regions, and similar humid regions, where actual water loss is mainly controlled by atmospheric demand.
Future water management must also consider the recent shifts of climatic control on the hydrological cycles. Because atmospheric demand (VPD) is a major control on potential water loss over the study region, predicting hydrological change under a changing climate must consider both air humidity and air temperature. Climate predictions from General Circulation Models (GCMs) must be assessed for their accuracy to simulate VPD in addition to air temperature and precipitation. In addition, potential ET algorithms that are often embedded in watershed hydrological models must include VPD as a major input variable to fully account for atmospheric water demand and actual ET.

Please also note the supplement to this comment:
https://www.hydrol-earth-syst-sci-discuss.net/hess-2017-241/hess-2017-241-AC2-supplement.pdf